# Ginseng Extract Ameliorates the Negative Physiological Effects of Heat Stress by Supporting Heat Shock Response and Improving Intestinal Barrier Integrity: Evidence from Studies with Heat-Stressed Caco-2 Cells, *C. elegans* and Growing Broilers

**DOI:** 10.3390/molecules25040835

**Published:** 2020-02-14

**Authors:** Georg Sandner, Andreas S. Mueller, Xiaodan Zhou, Verena Stadlbauer, Bettina Schwarzinger, Clemens Schwarzinger, Uwe Wenzel, Klaus Maenner, Jan Dirk van der Klis, Stefan Hirtenlehner, Tobias Aumiller, Julian Weghuber

**Affiliations:** 1School of Engineering and Environmental Sciences, University of Applied Sciences Upper Austria, Stelzhamerstraße 23, Wels 4600, Austria; georg.sandner@fh-wels.at (G.S.); verena.stadlbauer@ffoqsi.at (V.S.); bettina.schwarzinger@fh-wels.at (B.S.); 2Delacon Biotechnik GmbH, Weissenwolffstraße 14, Steyregg 4221, Austria; emily.zhou@delacon.com (X.Z.); jandirk.vanderklis@delacon.com (J.D.v.d.K.); hirtenlehner@agromed.at (S.H.); tobias.aumiller@delacon.com (T.A.); 3FFoQSI GmbH-Austrian Competence Centre for Feed and Food Quality, Safety and Innovation, Technopark 1C, Tulln 3430, Austria; 4Johannes Kepler University, Institute for Chemical Technology of Organic Materials, Linz, Austria 4040; clemens.schwarzinger@jku.at; 5Molecular Nutrition Research, Interdisciplinary Research Centre, Justus-Liebig-University of Giessen, Heinrich-Buff-Ring 26-32, 35392 Giessen, Germany; uwe.wenzel@ernaehrung.uni-giessen.de; 6Institute of Animal Nutrition of Free University Berlin, Königin-Luise-Str.49, 14195 Berlin, Germany; katmaenn@zedat.fu-berlin.de

**Keywords:** Ginseng extract, heat stress, intestinal barrier, broiler

## Abstract

Climatic changes and heat stress have become a great challenge in the livestock industry, negatively affecting, in particular, poultry feed intake and intestinal barrier malfunction. Recently, phytogenic feed additives were applied to reduce heat stress effects on animal farming. Here, we investigated the effects of ginseng extract using various in vitro and in vivo experiments. Quantitative real-time PCR, transepithelial electrical resistance measurements and survival assays under heat stress conditions were carried out in various model systems, including Caco-2 cells, *Caenorhabditis elegans* and *jejunum* samples of broilers. Under heat stress conditions, ginseng treatment lowered the expression of *HSPA1A* (Caco-2) and the heat shock protein genes *hsp-1* and *hsp-16.2* (both in *C. elegans*), while all three of the tested genes encoding tight junction proteins, *CLDN3*, *OCLN* and *CLDN1* (Caco-2), were upregulated. In addition, we observed prolonged survival under heat stress in *Caenorhabditis elegans*, and a better performance of growing ginseng-fed broilers by the increased gene expression of selected heat shock and tight junction proteins. The presence of ginseng extract resulted in a reduced decrease in transepithelial resistance under heat shock conditions. Finally, LC-MS analysis was performed to quantitate the most prominent ginsenosides in the extract used for this study, being Re, Rg1, Rc, Rb2 and Rd. In conclusion, ginseng extract was found to be a suitable feed additive in animal nutrition to reduce the negative physiological effects caused by heat stress.

## 1. Introduction

Global warming and the growing consumer demand for animal-derived food and meat, particularly in newly industrializing countries with hot climates, have increased public concerns regarding animal welfare, long-term sustainable livestock production and food and consumer safety [1,2].

Heat stress is an important environmental challenge for the livestock industry, representing the reaction of animals to high-temperature and high-humidity environments, and additionally produces unfavorable consequences, ranging from discomfort to death [3]. Under heat stress, animals change their behavior and physiology for body heat dissipation; they spend less time feeding and moving and more time resting, drinking and panting. The importance of animal responses to environmental challenges applies to all species. However, poultry are more susceptible to heat stress than other farm animals due to their fast metabolic rates [4].

Obvious signs of heat stress in poultry include poor production performance (e.g., decreased feed intake and daily weight gain), poor meat and egg quality, and high morbidity and mortality rates [5,6,7].

Moreover, heat stress impairs intestinal barrier function and induces immunosuppression, which in turn increases susceptibility for infectious diseases. The intestinal barrier is the first line of defense against harmful microbial pathogens and antigens from the intestinal lumen. Additionally, the intestinal barrier is formed by a layer of epithelial cells and sealed by tight junctions, which are mainly composed of transmembrane proteins such as claudins and occludins [8]. Tight junctions act as selective barriers that regulate paracellular transport. Heat stress can compromise the integrity of the tight junctions and thereby increase the permeability of the intestinal mucosa, disturbing its function as a selective barrier that absorbs nutrients while keeping pathogens at bay [9]. Heat stress was also reported to reduce the performance of broilers, increase viable counts of coliforms and Clostridium, downregulate the protein levels of occludin and zonula occludens-1, increase intestinal permeability and injure jejunal morphology [10].

Under heat stress, the level of reactive oxygen species in animals is also increased. Consequently, the body enters a stage of oxidative stress and starts producing and releasing heat shock proteins. Heat shock proteins constitute a cytoprotective system with a wide range of functions. These proteins are essential for the anti-inflammatory response and antigen presentation and are targets for antibodies, which activate the immune system and upregulate cellular immunity [11]. The modulation of heat shock proteins significantly protected the integrity of the intestinal mucosa of heat-stressed broilers, elevated antioxidant enzyme activities and relieved and reduced intestinal mucosal oxidative injury by inhibiting lipid peroxidation [12]. In contrast, the treatment of birds with inhibitors of heat shock proteins completely abolished these protective effects under heat stress [12]. In addition to traditional environmental management approaches, dietary supplements effectively counteract heat stress. For example, betaine is widely used in farm animals to combat heat stress [13,14,15,16].

Ginseng is a highly regarded herbal medicine in eastern Asia that exerts positive health effects by combating stress, enhancing immunity and promoting regeneration [17]. Ginseng extract consists of characteristic ginsenosides, a class of natural triterpene saponins. Numerous studies have shown that ginsenosides have multiple functions, including antioxidant enzyme modulation and intestinal barrier protection [18,19,20]. For instance, ginsenoside Rb2 induces superoxide dismutase (SOD) transcription through the activation of the SOD gene promoter region [19]. Ginsenoside of wild ginseng prevents the decrease in zonula occludens-1 in colonic epithelial cells from a murine colitis model, suggesting that ginseng is effective in the production and maintenance of tight junctions and in improving intestinal barrier function [20]. Kim and co-workers observed that red ginseng administered to rats during heat stress drastically suppressed lipid peroxidation and ROS-associated gene expression [21]. In addition, red ginseng decreased the gene expression of heat shock protein 70 without alleviating the weight loss of rats under heat stress [21]. To the best of our knowledge, only a few systematic studies are available on the effects of ginseng extract on animal performance under heat stress, on heat shock protein response and on intestinal barrier integrity.

Thus, the present study, based on in vitro and in vivo experiments with Caco-2 cells, the nematode *Caenorhabditis elegans* and growing broilers, aims to investigate the effects of ginseng extract on the parameters of heat shock protein response and intestinal barrier function. *C. elegans* was used as a convenient model to study the effects of ginseng extract on crucial mechanisms known to be involved in the hormetic stress response, such as the nuclear translocation of the forkhead box protein O (FOXO) transcription factor daf-16 [22]. Caco-2 intestinal epithelial cells were used for studies on intestinal barrier function due to their similarities to mammalian intestinal epithelia regarding transepithelial electrical resistance (TEER) [23]. Growing broilers were finally used to investigate whether the effects measured in *C. elegans* and Caco-2 cells correlated with effects on heat stress response and health in farm animals.

## 2. Results

### 2.1. Study With Caco-2 Cells

To investigate the influence of ginseng extract on the heat shock response in intestinal epithelial cells, the dose-response relationship between ginseng extract (0, 22.5, 45, 90, and 135 mg/L) and the mRNA concentrations of the known heat shock protein genes *HSPA1A* and *HSPB1* and the tight junction protein genes *CLDN1*, *CLDN3* and *OCLN* in Caco-2 cells were studied. As shown in Figure 1, compared with the control condition at 37 °C, heat stress at 41 °C for 3 h significantly increased *HSPA1A*, *HSPD1* and *HSPB1* mRNA concentrations. The cells had a 38.1-fold higher *HSPA1A*, 1.4-fold higher *HSPD1* and 2.9-fold higher *HSPB1* mRNA levels under heat stress than under normal temperature conditions. This finding indicates that incubating cells at 41 °C for 3 h produced sufficient heat stress in Caco-2 cells. Treatment with ginseng extract tended to dose-dependently lower the increase in *HSPA1A* mRNA concentration under heat stress conditions, and the effect was significant at the highest concentration (Figure 1A). The lowest and highest concentrations of ginseng extract (22.5 and 135 mg/L) decreased *HSPA1A* mRNA expression by 5.1-fold and 10.5-fold, respectively. In contrast, for *HSPD1* and *HSPB1*, the response to heat stress was increased by treatment with ginseng extract (Figure 1B,C). With one exception, all concentrations had significant effects on both genes.

Heat stress conditions significantly decreased the mRNA concentration of the tight junction protein genes *CLDN3* and *OCLN* (Figure 2B,C). The mRNA concentration after heat stress was only 70.1% and 68.6% of that under normal temperature conditions for *CLDN3* and *OCLN*, respectively. However, the mRNA expression of *CLDN1* was not significantly influenced by heat stress at 41 °C for 3 h (Figure 2A). Compared with the heat stress control condition without ginseng treatment, treatment with ginseng extract upregulated all three tight junction protein genes, though the effect was not dose dependent and not significant, except for the effect of one concentration on *OCLN*. For *CLDN1*, a higher expression level was detected in cells treated with ginseng than in cells cultivated at the normal temperature. The decreases in *CLDN3* and *OCLN* mRNA expression due to heat stress were reduced by treatment with ginseng extract. Notably, ginseng extract with a ginsenoside concentration of 90 mg/L had the strongest effect on all three tight junction protein genes. Expression levels of *CLDN1*, *CLDN3* and *OCLN* were increased by 27.5%, 13.5% and 30.2%, respectively, compared to control cells under heat stress conditions.

Finally, we tested the potential positive effect of ginseng extract on intestinal barrier integrity. As shown in Figure 3, ginseng extract applied at two different concentrations (90 and 135 mg/L) resulted in a reduced decrease in TEER values under heat stress conditions (45 °C).

### 2.2. Study With C. elegans

The influence of ginseng extract on heat shock response (*hsp-1*, *hsp-16.2*) and oxidative stress (*daf-16*) was tested in *C. elegans*. As shown in Figure 4, gene expression was increased in the group treated with heat stress for 1 h at 37 °C compared to the control group, indicating a sufficient heat stress reaction in *C. elegans*. Ginseng treatment (135 mg/L ginseng extract) lowered the induction of the selected genes by heat stress. *hsp-1* and *daf-16* showed a significant reduction in relative gene expression, while *hsp-16.2* showed a decreasing trend.

The incubation of wildtype nematodes in liquid medium containing 85 mg/L of water-soluble ginseng extract in addition to *E. coli* bacteria led to a significant increase in survival time under heat stress (Figure 5A). This increased resistance at 37 °C was associated with the nuclear localization of the FOXO transcription factor *daf-16* (Figure 5B). The fact that the activity of *daf-16* is required for the extended survival time caused by ginseng extract was finally proven by the RNAi-mediated knockdown of *daf-16*, which completely prevented the ginseng-induced increase in survival time (Figure 5C).

### 2.3. Study With Growing Broilers

The growth performance parameters of broiler chickens are presented in Table 1. The results of this study indicate a significant (*p* = 0.034) 5.2% improvement in feed conversion ratio (FCR) (1–42 d) in the ginseng extract group compared with the negative control group. Before the onset of heat stress (1–21 d), neither the positive control (betaine) nor the supplementation of ginseng extract produced significant differences in feed intake, body weight on day 21, and body weight gain from the negative control group. However, a positive influence on body weight gain (+5.1%) was observed in broilers fed the ginseng extract compared to negative control broilers. FCR was improved by approximately 6.1% (*p* = 0.074) in broiler chickens fed the ginseng extract compared to negative control broilers. From day 22 to day 42, during the heat stress challenge, the body weight gain in the ginseng extract group was 5.1% higher than that in the negative control group. In addition, the body weight gain in the betaine group (positive control) was 5.0% higher than that in the negative control group. Under heat stress, the feed intake of chickens fed diets containing ginseng extract was similar to that of the negative control chickens. Compared with the negative control chickens under heat stress, broiler chickens fed diets supplemented with ginseng extract showed a 4.9% improvement in FCR.

The influence of ginseng extract on heat shock response (*HSPA2*) and intestinal tight junction parameters (*CLDN1* and *OCLN*) was tested in 42-d-old broilers. As shown in Figure 6, compared with the non-treated control group, the group exposed to cyclic heat stress over 20 d with an average temperature of 34 °C showed increased gene expression, indicating a sufficient heat stress reaction. The ginseng treatment (90 mg/L ginseng extract) tended to increase heat stress induction, similar to treatment with ActiBeet^®^, a natural betaine source that served as an internal positive control [24,25]. The tight junction proteins showed increased relative gene expression in the ginseng-treated group compared to the non-treated control group; *CLDN1* tended to increase, while *OCLN* was significantly increased.

### 2.4. HPLC Analysis of Ginseng Extract

In order to identify the main ginsenosides in the used ginseng extract, HPLC-MS was applied (Figure 7). We quantitated twelve ginsenosides (nR1, Re, Rg1, Rf, Rb1, Rg2, Rh1, Rc, Rb2, Rd, CK, Rh2) with Re, Rd, Rc and Rb2 being the most abundant ones (Table 2). In addition, seven ginsenosides (gRf, F5, D3, Rb3, F1, F2 and Rg3) were identified without quantitation based on the *m/z* value and the retention time. In total, 384 mg g^−1^ of the 80% ginseng extract were quantitated.

## 3. Discussion

Heat stress profoundly affects the overall physiology, health, and productivity of farm animals. Under heat stress, the intestinal barrier function of animals is compromised, resulting in decreased production performance and increased morbidity and mortality [27]. Ginseng extract is the most popular medicinal herb in the Asian-Pacific region. Ginseng contains high levels of bioactive ingredients, such as acidic polysaccharides, saponins, and ginsenosides. Moreover, ginseng is regarded as a tonic with stimulant and stress-combating properties [17,28].

In this study, ginseng extract showed a beneficial influence in vitro, on the performance of growing broilers and on the heat resistance of *C. elegans*. The presented results for growing broilers show that, compared with no supplementation and the betaine positive control, ginseng extract improved body weight gain by ameliorating FCR under heat stress. Previous results describing the effect of ginseng extract on laying hens and weaned piglets are consistent with our findings [29,30]. Dietary supplementation of a red ginseng by-product significantly increased hen-day egg production, numerically increased egg weight and improved FCR [29]. Yang et al. reported significantly higher final body weight, a better feed utilization rate, and a lower incidence of diarrhea in weaned piglets fed ginseng polysaccharides than in control piglets [30].

In the current study, *C. elegans* was used as a traceable model to investigate the molecular pathways induced by ginseng extract during heat stress exposure in vivo and to explore possible ginseng-mediated adaptation mechanisms. Ginseng extract clearly conferred protection to wildtype nematodes under heat stress, resulting in lifespan extension. This effect was accompanied by a distinct increase in the nuclear translocation of the transcription factor *daf-16*, which is known to control the transcription of classic stress response genes, including those encoding small heat shock proteins such as *HSPB1* or antioxidant enzymes such as manganese superoxide dismutase-3 [31]. Our data suggest that the lifespan extension induced by feeding ginseng extract resulted from the transcriptional activation of protective target genes by *daf-16*. This hypothesis was shown by the knockdown of *daf-16* using RNAi, resulting in the complete abolishment of the lifespan extension induced by incubation with ginseng extract in *C. elegans*. Thus, it can be assumed that *daf-16* induces the transcription of several genes encoding cytoprotective and antioxidant proteins [32].

Heat shock response in this study was demonstrated using gene expression analysis of the heat shock protein *HSPB1* (Caco-2; *C. elegans*: *hsp-16.2*) and the HSP70 family (Caco-2: *HSPA1A*; *C. elegans*: *hsp-1*; broiler: *HSPA2*). These two heat shock proteins are heat-inducible molecular chaperones pivotal for the maintenance of cell homeostasis by controlling protein synthesis, folding, trafficking, aggregation, disaggregation, and degradation [33]. Heat exposure has been reported to inhibit protein folding, resulting in the accumulation of denatured and misfolded proteins [34]. *HSPB1* can bind denatured proteins in an ATP-independent manner, preventing their irreversible aggregation, but is unable to restore misfolded proteins; instead, it can transfer them to ATP-dependent *HSP70*, which promotes protein refolding [33]. Although *HSP70* gene expression was not affected by ginseng extract in broiler chickens, we observed that the mRNA levels of *HSPB1* in Caco-2 cells increased, while *HSP70* decreased following the administration of ginseng extract in heat-stressed *C. elegans* and in Caco-2 cells. The differential mRNA expression patterns of *HSPB1* and *HSP70* indicate that ginseng extract helps these organisms combat heat stress, mainly in an ATP-independent manner, which avoids spending excess ATP on survival and may shift energy flux to growth performance. Consistent with our findings, several studies have shown that ginseng extract inhibited the expression of *HSP70* (mRNA and protein levels) in heat-stressed rats [21,35]. In contrast, Yeo et al. reported that ginseng extract has a cytoprotective effect on ethanol-induced gastric damage through the induction of *HSP70* [36]. This discrepancy could be explained by the use of different stress models. To the best of our knowledge, the expression of *HSPB1* under heat stress conditions and ginseng extract treatment has not yet been described. Similar to our results, the pretreatment of rats with ginseng extract ameliorated pathological changes in the gastric mucosa by inducing the overexpression of *HSPB1* [36].

The intestinal barrier is formed by enterocyte membranes and tight junctions in the intestinal epithelium. Heat stress has been shown to disrupt the integrity of the intestinal barrier by damaging tight junction proteins. Claudins and occludins, two classes of important tight junction protein, act as gate guards and border protectors that regulate paracellular transport. It is well documented that intestinal barrier dysfunction is related to the loss of tight junction proteins, and both claudin-1 and occludin are crucial for tightening and stabilizing the barrier in epithelial cells [10,37,38]. Our results demonstrate that ginseng extract significantly elevated the mRNA abundance of *CLDN1* (encoding claudin-1) and *OCLN* (encoding occludin) in heat-stressed broiler chickens. This finding is in accordance with our findings in Caco-2 cells, in which *CLDN1* and *OCLN* mRNA levels increased after heat exposure. Ginseng extract enhanced the expression of tight junction proteins, suggesting its ability to improve intestinal barrier integrity under heat stress. The effect of ginseng extract on other stress models concurs with our findings. Ginseng saponins have been reported to increase the abundance of the tight junction proteins claudin-1 and occludin in ageing rats [39]. Ginseng oligopeptides have been shown to alleviate irradiation-induced intestinal injury in mice and Caco-2 cells by promoting occludin and *ZO-1* expression, maintaining intestinal barrier integrity, and resulting in increased transepithelial electrical resistance compared to non-treated controls [40]. The exact mechanisms by which ginseng extract modulates tight junction proteins are unknown, but there are several plausible explanations. First, ginseng extract may influence tight junction protein activity via heat shock proteins. The heat shock protein Apg-2, a molecular chaperone of the HSP70 family, has been reported to competitively bind to and thereby inhibit signal transduction activity mediated by the tight junction protein *ZO-1* [41]. Second, ginsenosides, the main active compound of ginseng extract, may act as agonists or antagonists of nuclear transcription factors, governing the gene expression of tight junction proteins, since direct interactions of ginsenosides with steroid hormone receptors and the indirect modulation of receptor tyrosine kinases, serotonin receptors, and NMDA receptors are well documented [28]. Third, ginseng extract may regulate the expression of tight junction proteins via cytokines. For instance, ginseng extract has an anti-inflammatory effect by inhibiting NF-κB, consequently decreasing IL-6 in various stress models (heat stress and colitis models) [20,35]. IL-6 regulates the expression of the tight junction protein claudin-2, which forms a paracellular channel for small cations and water [42].

Our TEER measurements indicate a connection between the expression of tight junction proteins and the integrity of the intestinal barrier. The decrease in TEER values due to induced heat stress was reduced by treatment with ginseng extract. Thus, the intestinal barrier was improved, as it was more resistant to heat stress.

We used HPLC-MS analysis to identify the main ginsenosides present in the 80% ginseng extract applied in the in vitro and in vivo experiments. The amount and prevalence of the identified compounds are in agreement with other studies analyzing extracts from various ginseng sources [43,44]. The most abundant ginsenosides present in our extract were Re, Rg1, Rc, Rb2 and Rd. Re and Rb2 have previously been reported to exert anti-oxidative and anti-inflammatory activities, confirming their potential bioactivity, which might be relevant for the effects described in this study [44,45].

In conclusion, the present study clearly shows that ginseng extract increases the performance of heat-stressed broiler chickens and the heat resistance of *C. elegans* by modulating the gene expression of heat shock proteins and tight junction proteins and by promoting nuclear *daf-16* translocation in vivo (broilers and *C. elegans*) and in vitro (Caco-2 cells). These results suggest that ginseng extract mitigates the adverse effects of heat stress by supporting the heat shock response and improving the intestinal barrier integrity of animals. Accordingly, we conclude that ginseng extract can be a useful feed ingredient to maintain the health and productivity of farm animals under environmental heat stress conditions.

## 4. Materials and Methods

### 4.1. Study With Caco-2 Cells

#### 4.1.1. Caco-2 Cell Culture and Differentiation

Human Caco-2 cells (DSMZ, Braunschweig, Germany) were maintained in MEM with Earle’s salts supplemented with 10% FBS and 100 U/mL penicillin/100 µg/mL streptomycin (Biochrom GmbH, Berlin, Germany) and grown at 37 °C in a humidified atmosphere (≥95%) with 5% CO_2_. For qPCR experiments, the cells were seeded in 12-well plates at 1.2 × 10^6^ cells per well. For the TEER measurements in transwell inserts (8.4 mm, collagen-treated, 0.4 µm pore diameter), the cells were seeded at 1.65 × 10^5^ per insert to reach confluency on the next day (Greiner Bio-One International GmbH, Kremsmünster, Austria). The cells were further maintained in Entero-STIM Intestinal Epithelium Differentiation Medium supplemented with 0.1% MITO+ Serum Extender (Corning, Wiesbaden, Germany) and 100 U/mL penicillin/100 µg/mL streptomycin, and the medium was changed daily. The qPCR experiment was carried out on day 5, when the cells were completely differentiated. The measurement of intestinal barrier integrity was performed on day 7, when TEER reached values of at least 500 Ω.

#### 4.1.2. Induction of Heat Stress

To analyse the influence of ginseng extract (ginseng extract, 80% ginsenosides, Denk Ingredients, article number 966208, Munich, Germany) on the expression of different genes via qPCR, the extract was added to the cells on day 4, and the cells were incubated with the extract overnight for 15 h. The extract was dissolved in complete differentiation medium and diluted to final ginsenoside concentrations of 22.5, 45, 90 and 135 mg/L. To induce heat stress, the samples were incubated at 41 °C for 3 h, while control samples were incubated at 37 °C. For TEER measurements, the cells were treated with ginseng extract throughout the whole differentiation period, starting on day 2. Ginsenoside concentrations of 90 and 135 mg/L were used, and the solution was changed on day 5 and on day 7, shortly before starting TEER measurements. Heat stress was induced at 45 °C over a period of 5 h.

#### 4.1.3. Detection of *HSPA1A*, *HSPD1*, *HSPB1*, *CLDN1*, *CLDN3* and *OCLN* mRNA Expression by Quantitative Real-Time PCR

The mRNA expression of heat shock protein family A (HSP70) member 1A (*HSPA1A*), heat shock protein family D (Hsp60) member 1 (*HSPD1*), heat shock protein family B (small) member 1 (*HSPB1*), claudin-1 (*CLDN1*), claudin-3 (*CLDN3*) and occludin (*OCLN*) was measured quantitatively by real-time PCR (C1000 Thermal Cycler and CFX96 Real-Time System, Bio-Rad Laboratories, Vienna, Austria). Total RNA was isolated with the RNeasy Plus Mini Kit (Qiagen, Hilden, Germany) followed by the measurement of the RNA integrity with a BioAnalyzer 2100 and RNA 6000 Nano Kit (both Agilent Technologies, Santa Clara, USA). Only RNA samples with RIN > 7 were used for further analysis. Next, 500 ng of total RNA was transcribed into cDNA using the iScript cDNA Synthesis Kit (end volume: 20 µL), and real-time PCR with the iQ SYBR Green Supermix was carried out according to the manufacturer´s instructions (both from Bio-Rad Laboratories, Vienna, Austria). For real-time PCR, 2 µL of cDNA was added to 18 µL of master mix (10 µL of iQ SYBR Green Supermix (2×), 2 µL of primer [3 pmol/µL of each the forward and reverse primer], 6 µL of nuclease-free water). DNA denaturation and polymerase activation were performed for 3 min at 95 °C, followed by 40 PCR cycles. One amplification cycle was divided into three parts: denaturation at 95 °C for 10 s, annealing and extension at 60 °C for 30 s, and a plate read after each cycle. Finally, melt curve analysis was performed by gradually increasing the temperature to 95 °C to exclude the formation of primer dimers. Agarose gel electrophoresis was carried out to exclude unspecific products. The mRNA expression of the following six different reference genes was analysed in each experiment: glyceraldehyde-3-phosphate dehydrogenase (*GAPDH*), beta actin (*ACTB*), ribosomal protein L5 (*RPL5*), ribosomal protein lateral stalk subunit P0 (*RPLP0*), hypoxanthine phosphoribosyltransferase 1 (*HPRT1*) and beta-2-microglobulin (*B2M*). In each experiment, the three most stable reference genes were determined according to Vandesompele et al. [46]. The primers used for amplification are listed in Table 3 (Eurofins Genomics, Ebersberg, Germany or Microsynth AG, Balgach, Switzerland).

#### 4.1.4. Validation of Intestinal Barrier Integrity by TEER Measurements

Intestinal barrier integrity was analysed by a transwell insert-based assay with Caco-2 cells, as described previously [47,48]. The TEER values were measured each day during differentiation with a Millicell-ERS-2 Voltohmmeter (Merck KGaA, Darmstadt, Germany). When the values reached levels higher than 500 Ω on day 7, the assay was started. After changing the medium, TEER values were determined in the presence or absence of ginseng extract. Afterwards, heat stress was induced in all groups except for the control group, and the TEER values were recorded over 5 h after 1, 2, 3.5 and 5 h.

### 4.2. Study With C. elegans

#### 4.2.1. *C. elegans* Maintenance

*C. elegans* wildtype strain N2, variation Bristol, and strain TJ356 *daf-16::gfp* (zls356) were obtained from the *C. elegans* Genetics Center (CGC; University of Minnesota, MN, USA). The nematodes were maintained on nematode growth medium (NGM) agar plates seeded with *E. coli* OP50 at 20 °C according to standard protocols [53]. *E. coli* HT115 RNAi clones were purchased from Source Bioscience (Cambridge, UK) and included a negative control (empty L4440 vector) and daf-16 (R13H8.1). Methods such as freezing nematodes and obtaining synchronous populations using bleach with the hypochlorite treatment of egg-laying adults were also performed according to standard protocols [53].

#### 4.2.2. Treatment of Nematodes with Ginseng Extract

For the qPCR experiments, synchronized N2 worms were seeded on 6 cm NGM plates containing only OP50 (control) or OP50 with ginseng extract (135 mg/L final concentration). The NGM plates of the treatment group also contained ginseng extract at the same concentration. For the thermotolerance assay, synchronous nematodes were raised in liquid culture using NGM liquid and packed *E. coli* HT115 according to Stiernagle [53]. Carbenicillin was added to the NGM liquid to inactivate *E. coli*. A volume of 56 µL of NGM liquid was dispensed into each well of a 96-well microplate, to which 10 µL of M9 buffer containing 10 synchronized L1 larvae was added. L1 larvae were maintained shaking at 20 °C and reached the adult stage within 3 d. Ginseng extracts were prepared as 425, 850 and 1700 mg/L stock solutions in M9 buffer and sonicated for 5 min. Seven microliters of each plant extract stock solution was added to the incubation medium to reach the final concentrations of 42.5, 85 and 170 mg/L. The control nematodes were always treated with identical volumes of M9 buffer instead of ginseng.

#### 4.2.3. mRNA Expression of *hsp-16.2*, *hsp-1* and *daf-16* by Quantitative Real-Time PCR

The mRNA expression levels of the heat shock protein Hsp-16.2 (*hsp-16.2*), heat shock 70 kDa (HSP70) protein A (*hsp-1*) and the FOXO transcription factor (*daf-16*) were measured quantitatively by real-time PCR (C1000 Thermal Cycler and CFX96 Real-Time System, Bio-Rad Laboratories, Vienna, Austria). Seventy-two hours after seeding, the adult worms were heat-shocked for 1 h at 37 °C, while the control group was kept at the maintenance temperature of 20 °C. RNA was then isolated via TRIzol-chloroform (ThermoFisher Scientific, Vienna, Austria) extraction followed by gDNA removal using the TURBO DNA-free Kit (ThermoFisher Scientific, Vienna, Austria) according to the manufacturer’s instructions. cDNA synthesis and qPCR were carried out as stated in the cell culture section, but with the following minor modifications: 50 ng of purified RNA was transcribed into cDNA, and for qPCR, annealing at 57.5 °C and extension at 72 °C for 20 s were selected. The gene expression of the target genes in each experiment was normalized to the expression of multiple reference genes, namely, beta Actin (*act-1*), DNA-directed RNA polymerase II subunit RPB1 (*ama-1*) and Peroxisomal Membrane Protein-related protein (*pmp-3*). The oligonucleotide sequences of the primers used are shown in Table 3.

#### 4.2.4. Determination of Survival Under Heat Stress

After incubating young adult N2 nematodes for 48 h at 20 °C in the presence or absence of ginseng extract, their survival was determined using a microplate thermotolerance assay as described in [54]. Briefly, nematodes were washed out of the wells with M9 buffer/Tween20 (1% *v*/*v*) into 15-mL tubes, followed by an additional three washing steps. In each well of a black 384-well low-volume microtiter plate, 6.5 μL of M9 buffer/Tween20 (1% *v*/*v*) solution was added. Subsequently, one nematode was dispensed in 1 μL of M9 buffer under a stereomicroscope (Breukhoven Microscope Systems) into each well and mixed with 7.5 μL of SYTOX green to reach a final concentration of 1 μM. To prevent water evaporation, the plates were sealed with Rotilab sealing film and covered with a lid. Heat shock (37 °C) was induced, and fluorescence was measured with a Fluoroskan Ascent microtiter plate reader (Thermo Labsystems, Bonn, Germany) every 30 min. To detect SYTOX green fluorescence, the excitation wavelength was set to 485 nm, and emission was measured at 538 nm. To determine the survival time for each nematode, an individual fluorescence curve was generated. The time of death was defined as one hour after an increase in fluorescence over the baseline level was observed and was first verified by touch provocation. Based on the individual times of death, Kaplan–Meier survival curves were drawn.

#### 4.2.5. Nuclear *daf-16* Translocation

*Daf-16* localization in the transgenic strain TJ356, which expresses a fusion protein of *daf-16* and green fluorescent protein (GFP), was accomplished by using an EVOS LED Light Cube GFP, with an excitation of 470 nm and an emission of 525 nm after incubating the nematodes for 48 h in the absence or presence of ginseng extract as described above. The worms were anaesthetised by the addition of 2 mM levamisole. Distinct fluorescence spots were regarded as nuclear localization, whereas diffuse fluorescence was characterized as cytosolic localization. Images were collected at tenfold magnification, and the quantification of the different fluorescence patterns was performed using ImageJ (National Institute of Health, NIH).

#### 4.2.6. *Daf-16* RNAi

RNAi experiments were performed in liquid cultures as previously described [55,56]. In brief, to induce interference, the expression of gene-specific dsRNA in the corresponding RNAi feeding strain was induced with 1 mM isopropyl-β-d-thiogalactopyranoside for 1 h at 37 °C. Subsequently, bacterial cells were washed and resuspended in NGM that contained 50 µg/mL kanamycin to inactivate bacteria. A total of 44 μL of this suspension was dispensed into each well of a 96-well plate (Greiner Bio-One, Frickenhausen, Germany), to which 10–15 synchronized L1 larvae were added. In general, L1 larvae reached the adult stage within 3 d of incubation with agitation at 20 °C.

### 4.3. Study With Growing Broilers

#### 4.3.1. Animals, Diets and Experimental Design

The trial was performed in accordance with the Animal Welfare Act of Germany and was approved by the local state office of Occupational Health and Technical Safety (Landesamt für Gesundheit und Soziales, LaGeSo, no. A 0100/13). A total of 135 1-d-old healthy male broiler chickens (Cobb500), obtained from a local hatchery (Cobb Germany Avimex GmbH, Brösenweg 80, 04509 Wiesenena-Wiedemar), were used for the present study. The broiler barn has nine ground floor pens, and birds were randomly assigned to three experimental groups (negative control, positive control (betaine) and ginseng extract treatment) with three repetitions of 15 birds each to produce statistical relevant results, demanding a minimum of 3 repetitions per group. The average initial body weight of 43.3 ± 1.23 g was similar in all experimental groups. The 42-d feeding experiment consisted of a 21-d starter phase without heat stress, followed by a 21-d grower-finisher phase with cyclic heat stress. The experimental diets for both phases met the guidelines of the breeders’ nutritional recommendations for Cobb 500 broilers [57]. During the whole experiment, broilers were kept in floor pens with fresh wood shavings as bedding material and had ad libitum access to the mash feed and tap water supplied by drinking bowls. The diets were prepared to meet the macronutrient and micronutrient concentrations for growing broilers recommended by the Society of Nutrition Physiology and the National Research Council (Table 4). The diet of the negative control group contained no phytogenic feed additive. The diet of the positive control group was supplemented with 1 kg/t ActiBeet^®^ (Agrana, Tulln, Austria) containing 50% natural betaine from sugar beet. Regarding the diet of the treatment group, 112.5 g of the same ginseng extract as that used for the in vitro and *C. elegans* experiments was added, providing 90 g of pure ginsenosides. The premixes of all groups were filled with carrier (1:1 mixture of wheat semolina and wheat bran to the level of 5 kg/t). Consequently, the premix of the negative control contained only carrier. During the 21-d heat stress period, heat stress was applied from 9:00 to 17:00. The average barn temperature in this period was 34 °C. The cool night period with an average temperature of 26 °C lasted from 7.00 pm to 7.00 am. The lighting regime in the barn consisted of a 16-h light cycle at 60 lux and an 8-h dark cycle. The health status of the broilers was checked twice every day. The individual body weight of the broilers was measured every week. The weight of consumed feed was determined per experimental unit each week. From these weight data, the performance measures of feed intake, body weight gain and feed conversion ratio (FCR) were calculated each week. At the end of the trial, the birds that were further used for gene expression experiments were killed by decapitation after stunning for organ sampling. Jejunum samples were excised, snap frozen in liquid nitrogen and stored at −80 °C until quantitative gene expression analysis by real-time PCR. Broilers that were only part of the performance trial were not killed but used for commercial breeding instead.

#### 4.3.2. Detection of *HSP2A*, *CLDN1* and *OCLN* mRNA Expression by Quantitative Real-Time PCR

The mRNA expression levels of heat shock 70 kDa (HSP70) protein 2 (*HSP2A*) and the tight junction genes claudin-1 (*CLDN1*) and occludin (*OCLN*) in broilers were measured quantitatively by real-time PCR (C1000 Thermal Cycler and CFX96 Real-Time System, Bio-Rad Laboratories, Vienna, Austria). The jejunum samples of each broiler were homogenized in liquid nitrogen. The total RNA of a homogenized tissue aliquot was isolated with a Qiagen RNeasy Plus Mini Kit, followed by gDNA removal using a TURBO DNA-free Kit (Thermo Fisher Scientific) according to the manufacturer’s instructions. Next, cDNA synthesis and qPCR were carried out as stated in the cell culture section with the following minor modifications: 50 ng of purified RNA was transcribed into cDNA, and for qPCR, annealing at 57 °C and extension at 72 °C for 20 s were selected. The reference genes used were beta actin (*ACTB*), phosphoglycerate kinase 1 (*PGK1*), succinate dehydrogenase complex flavoprotein subunit A (*SDHA*) and tyrosine 3-monooxygenase/tryptophan 5-monooxygenase activation protein zeta (*YWHAZ*). The primer oligonucleotide sequences used for real-time PCR are listed in Table 3.

### 4.4. HPLC Analysis of Ginseng Extract

Ginsenosides standards were obtained from Extrasynthese (Lyon, France). The extract was analysed with HPLC-MS after a modified protocol described elsewhere [26,58,59]. The samples were dissolved in methanol and separated on a Surveyor HPLC (Thermo Fisher Scientific, Waltham, MA, USA) equipped with an Accucore C18 column (150 mm × 3 mm, 2.6 µm; Thermo Fisher Scientific, Waltham, MA, USA) using a gradient of 0.1% formic acid in water (A) and 0.1% formic acid in acetonitrile (B): from 80% A to 70% A in 5 min, to 55% A in 15 min, to 25% A in 10 min which was kept for 2 min. The column flow was 500 µL min^−1^ and the temperature was kept at 40 °C. A LTQ Orbitrap Velos (Thermo Fisher Scientific, Waltham, MA, USA) was used as a detector with an ESI ion source. The instrument was operated in FTMS negative mode with a resolution of 30,000 and a scan range from 200–2000 *m/z*. For quantitation, the extracted masses of the formiate adduct ions were used (see Table 2). Calibration was carried out with twelve ginsenoside standards in the range from 30 µg L^−1^ to 130 mg L^−1^ in 6 different concentrations, all measured in triplicate. The injection volume for all samples was 1 µL.

### 4.5. Calculations and Statistics

For gene expression studies and intestinal barrier integrity measurements, statistical analysis was carried out with GraphPad Prism version 6.07 (GraphPad Software, San Diego, CA, USA) using an unpaired t test with Welch’s correction. In Caco-2 qPCR experiments, the mean is based on six individual samples obtained on three different days measured in duplicate. The *C. elegans* qPCR experiments were also performed on three different days and measured in duplicate. The mean is based on a minimum of six individual samples. In the broiler qPCR experiments, six biological replicates (containing two jejunum samples) per treatment and two replicates of each sample were analysed by real-time PCR. In all real-time PCR analyses, the detected cT values were used to calculate the relative mRNA expression levels via the 2^−∆∆^*^CT^* method [60]. The gene expression of the target genes in each experiment was normalized to the expression of multiple reference genes. For intestinal barrier integrity measurements, the mean represents three individual inserts for each group. To compare the survival rates of *C. elegans*, a log rank test was performed. Kaplan-Meier survival curves are shown for the survival experiments. For comparisons among groups, analysis of variance (ANOVA) was performed. The differences between groups were determined by the Bonferroni–Holm multiple comparison test. In these studies, the results are presented as the mean ± SD with the significance levels of *p* < 0.05 (*), *p* < 0.01 (**), *p* < 0.001 (***) and *p* < 0.0001 (****). For the performance study of broilers, the statistical analyses were performed with the software package SPSS (IBM SPSS Version 21, IBM, Armonk, NY, USA) and were based on one-way ANOVA. All treatment least-square means were compared with each other and the Tukey adjustment was used to control for experiment-related error. The differences among least-square means with a probability of *p* ≤ 0.05 were accepted as statistically significant, and mean differences with *p*-values ranging from 0.06 to 0.10 were considered a trend.

## 5. Conclusions

In conclusion, a ginseng extract containing particularly the major ginsenosides Re, Rg1, Rc, Rb2 and Rd was found to be a suitable feed additive in animal nutrition to reduce the negative physiological effects caused by heat stress.

## Figures and Tables

**Figure 1 molecules-25-00835-f001:**
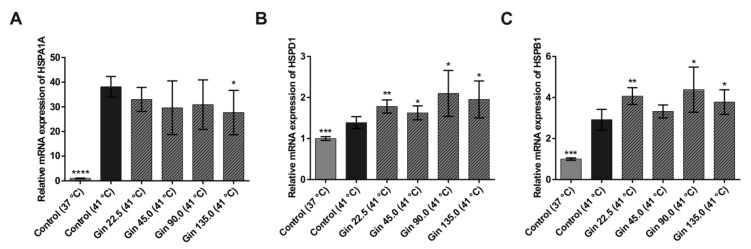
mRNA expression of heat shock protein genes in Caco-2 cells grown under normal temperature conditions (37 °C) or under heat stress (41 °C), without (control) or with the addition of ginseng extract. The final concentrations of ginsenosides were 22.5 mg/L (Gin 22.5), 45 mg/L (Gin 45.0), 90 mg/L (Gin 90.0) and 135 mg/L (Gin 135.0). The expression levels of *HSPA1A* (**A**), *HSPD1* (**B**) and *HSPB1* (**C**) are indicated as fold-changes normalized to the control at the normal temperature condition of 37 °C. Error bars are based on the standard deviation (*n* = 6). **** *p* < 0.0001, *** *p* < 0.001, ** *p* < 0.01 and * *p* < 0.05 indicate significant differences from the control under the 41 °C heat stress condition.

**Figure 2 molecules-25-00835-f002:**
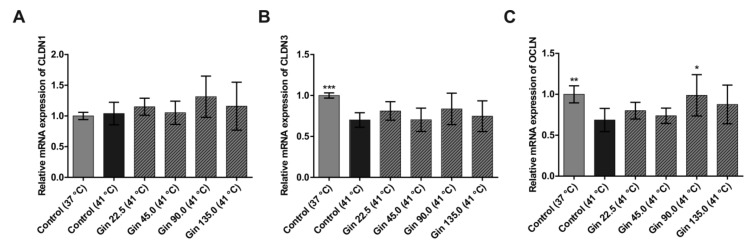
mRNA expression of tight junction protein genes in Caco-2 cells grown under normal temperature conditions (37 °C) or under heat stress (41 °C), without (Control) or with the addition of ginseng extract. The final concentrations of ginsenosides were 22.5 mg/L (Gin 22.5), 45 mg/L (Gin 45.0), 90 mg/L (Gin 90.0) and 135 mg/L (Gin 135.0). The expression levels of *CLDN1* (**A**), *CLDN3* (**B**) and *OCLN* (**C**) are indicated as fold-changes normalized to the control under the normal temperature condition of 37 °C. Error bars are based on the standard deviation (*n* = 6). *** *p* < 0.001, ** *p* < 0.01 and * *p* < 0.05 indicate significant differences from the control under the 41 °C heat stress conditions.

**Figure 3 molecules-25-00835-f003:**
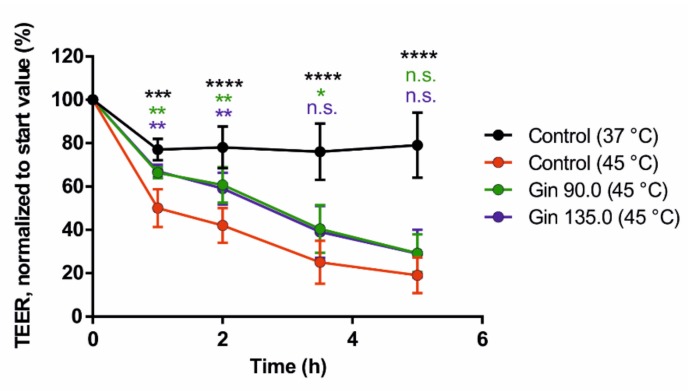
Effects of ginseng extract on the transepithelial electrical resistance (TEER) values of Caco-2 monolayers on transwell inserts, representing intestinal barrier integrity. Caco-2 cells were grown on collagen-coated 0.4-µm transwell inserts and differentiated into monolayers. The cells were incubated with ginseng extract at ginsenoside concentrations of 90 mg/L (Gin 90.0) and 135 mg/L (Gin 135.0) throughout the differentiation period. The TEER values are normalized to the respective start value. Error bars are based on the standard deviation (*n* = 6). **** *p* < 0.0001, *** *p* < 0.001, ** *p* < 0.01 and * *p* < 0.05 indicate significant differences from the control under the 45 °C heat stress condition.

**Figure 4 molecules-25-00835-f004:**
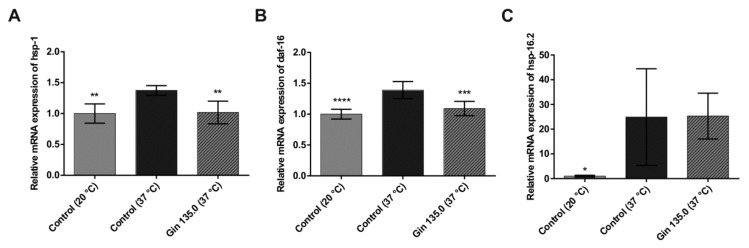
mRNA expression of heat shock protein genes and the oxidative stress-responsive FOXO gene in *C. elegans* grown under normal temperature conditions (20 °C) or under heat stress (37 °C) without (control) or with the addition of ginseng extract. The final concentration of ginsenosides was 135 mg/L (Gin 135.0). The expression levels of *hsp-1* (**A**), *daf-16* (**B**) and *hsp-16.2* (**C**) are indicated as the fold-change normalized to the control under the normal temperature condition of 20 °C. Error bars are based on the standard deviation (*n* = 6 (**A**), *n* = 9 (**B**,**C**)). **** *p* < 0.0001, *** *p* < 0.001, ** *p* < 0.01 and * *p* < 0.05 indicate significant differences from the control under 37 °C heat stress conditions.

**Figure 5 molecules-25-00835-f005:**
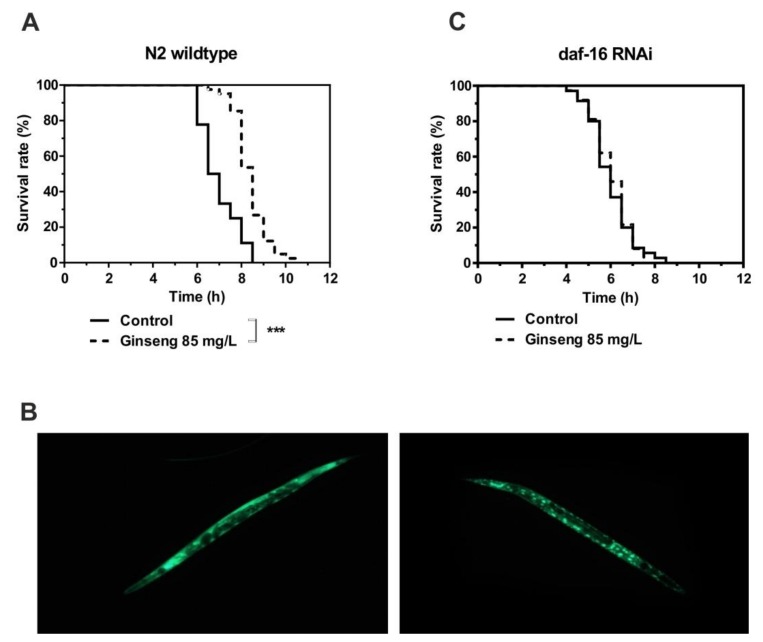
The survival time of N2 wildtype *C. elegans* at 37 °C in the absence (control) or presence of 85 mg/L ginseng extract was measured as described in the Methods section (**A**,**C**). Representative image of *daf-16::GFP* in strain TJ356 (**B**). *** *p* < 0.001 indicates significant differences from the control without ginseng extract treatment.

**Figure 6 molecules-25-00835-f006:**
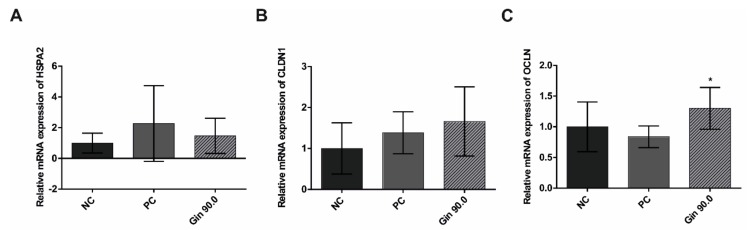
Jejunal mRNA expression of a heat shock protein gene and tight junction protein genes in growing broilers under heat stress conditions (day 22–42) fed diets containing no additive (negative control), 1000 mg/kg ActiBeet^®^, as natural betaine source (positive control), or 90 mg/kg ginsenosides (Gin 90.0). The expression levels of *HSPA2* (**A**), *CLDN1* (**B**) and *OCLN* (**C**) are indicated as fold-changes normalized to the average levels under the cyclic heat shock control temperature conditions of 34 °C. The error bars are based on the standard deviation (*n* = 6). * *p* < 0.05 indicates significant differences from the negative control under the 34 °C heat stress condition.

**Figure 7 molecules-25-00835-f007:**
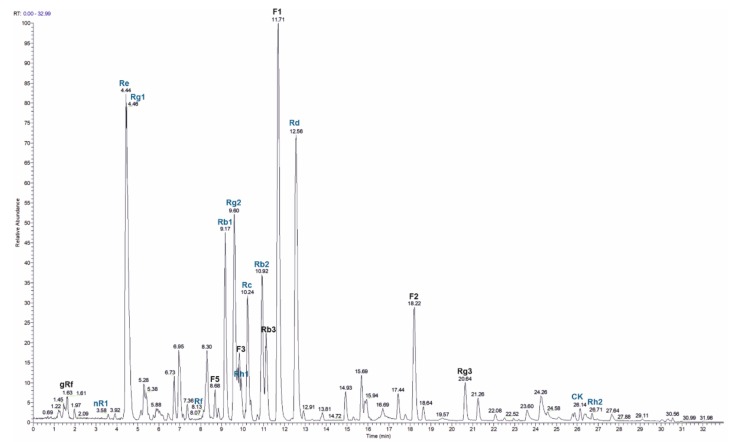
HPLC-MS analysis of 80% ginseng extract. A total ion current (TIC) chromatogram is shown, indicating seven ginsenosides (denoted in black) without and twelve ginsenosides (denoted in blue) with quantitation.

**Table 1 molecules-25-00835-t001:** Feed intake and growth performance of growing broilers under normal temperature conditions (days 1–21) and under heat stress conditions (days 22–42).

Items	Negative Control	Positive Control ActiBeet^®^	Ginseng Extract	*p*-Value
Feed intake (g), 1–21 d	866 ± 28.3	856 ± 27.5	856 ± 36.8	0.907
Body weight (g), 21 d	686 ± 35.7	690 ± 46.3	718 ± 17.7	0.519
Body weight gain (g), 1–21 d	643 ± 36.7	656 ± 33.4	676 ± 18.3	0.454
**Feed conversion (g/g), 1–21 d**	**1.350 ± 0.055 A**	**1.307 ± 0.027 AB**	**1.267 ± 0.021 B**	**0.086**
Feed intake (g), 22–42 d	1879 ± 39.6	1905 ± 42.5	1879 ± 100.2	0.869
Body weight (g), 42 d	1966 ± 74.9	2034 ± 48.0	2062 ± 58.1	0.221
Body weight gain (g), 22–42 d	1280 ± 64.8	1344 ± 94.3	1345 ± 63.6	0.527
**Feed conversion (g/g), 22–42 d**	**1.470 ± 0.080**	**1.421 ± 0.068**	**1.398 ± 0.017**	**0.395**
Feed intake (g), 1–42 d	2745 ± 61.7	2761 ± 16.1	2735 ± 79.6	0.867
Body weight gain (g), 1–42 d	1923 ± 74.9	1991 ± 47.0	2020 ± 58.0	0.214
**Feed conversion (g/g), 1–42 d**	**1.429 ± 0.038 a**	**1.387 ± 0.026 ab**	**1.354 ± 0.006 b**	**0.040**

Fed diets containing no additive (negative control), 1000 mg/kg ActiBeet^®^, as natural betaine source (positive control), or 90 mg/kg ginseng extract. ActiBeet^®^ and ginseng extract were applied throughout the trial (1–42 d). Capital letters (A and B) in a line indicate a statistic trend between groups (0.05 < *p* ≤ 0.10), whereas small letters in a line (a and b) indicate a significant difference between treatments (*p* ≤ 0.05).

**Table 2 molecules-25-00835-t002:** Identification and quantitation of ginsenosides.

Compound	*m/z* Observed	tR/min	c/mg.g^−1^	Standard Purity
20-O-glucoginsenoside gRf ^1^	1007.5455	1.63		
**Notoginsenoside nR1**	977.5354	3.7	1.1	≥98%
**Ginsenoside Re**	991.5505	4.35	143.4	≥98%
**Ginsenoside Rg1**	845.4929	4.44	35.2	≥98%
**Ginsenoside Rf**	845.4925	8.06	1.4	≥98%
Ginsenoside F5 ^1^	815.4811	9.15		
**Ginsenoside Rb1**	1153.6035	9.46	15.5	≥98%
**Ginsenoside Rg2**	829.4959	9.56	13.5	≥97%
Ginsenoside F3 ^1^	815.4818	9.59		
**Ginsenoside Rh1**	683.4388	9.73	6.1	≥98%
**Ginsenoside Rc**	1123.5927	10.12	40.2	≥98%
**Ginsenoside Rb2**	1123.5924	10.89	46.1	≥97%
Ginsenoside Rb3 ^1^	1123.5922	11.09		
Ginsenoside F1 ^1^	683.4383	11.68		
**Ginsenoside Rd**	991.5494	12.46	75.6	≥98%
Ginsenoside F2 ^1^	829.4966	18.19		
Ginsenoside Rg3 ^1^	829.4968	20.61		
**Ginsenoside CK**	667.4445	25.81	1.1	
**Ginsenoside Rh2**	667.4445	26.72	1.9	

^1^ Tentative identification based on the *m/z* value and retention time as described in Lee et al., 2017 [26]. Quantitated ginsenosides are highlighted in bold.

**Table 3 molecules-25-00835-t003:** Analysed genes in qPCR experiments with Caco-2 cells (human), *C. elegans* and broilers (*Gallus gallus*) and the oligonucleotide sequences of primers used.

Genes	Forward Primer Sequence (5’–3’)	Reverse Primer Sequence (5’–3’)	Accession No.
*Human*			
*ACTB* [49]	GCGGGAAATCGTGCGTGACATT	GATGGAGTTGAAGGTAGTTTCGTG	NM_001101
*B2M*	TGAAGCTGACAGCATTCG	CAGACACATAGCAATTCAGG	NM_004048
*CLDN1* [50]	AACGCGGGGCTGCAGCTGTTG	GATGTTGTCGCCGGCATA	NM_021101
*CLDN3*	CACGCGAGAAGAAGTACA	TCTGTCCCTTAGACGTAGT	NM_001306
*GAPDH*	TGGTATCGTGGAAGGACTCA	CAGTGAGCTTCCCGTTCAG	NM_002046
*HPRT1*	GACCCCACGAAGTGTTGGAT	ACTGGCGATGTCAATAGGACTC	NM_000194
*HSPA1A*	GTGGAGGAGTTCAAGAGAA	GGTGATGGACGTGTAGAA	NM_005345
*HSPB1*	CTGGATGTCAACCACTTCGC	TATTTCCGCGTGAAGCACC	NM_001540
*HSPD1*	GAAATTGCCAATGCTCACCG	CTTGACTGCCACAACCTGAA	NM_002156
*OCLN* [51]	GGACTCTACGTGGATCAGTATTTG	AATAATCATGAACCCCAGTACAATG	NM_002538
*RPL5*	TGGGCCAGAATGTTGCAGAT	AGGGACATTTTGGGACGGTT	NM_000969
*RPLP0*	TCTACAACCCTGAAGTGC	AAGGTGTAATCCGTCTCC	NM_001002
*C. elegans*			
*act-1*	TGTTCCCATCCATTGTC	GCTCATTGTAGAAGGTGTG	NM_073418
*ama-1*	CTCCGTCGTTGACTGTAT	ATACCCATTCCTCGTCTTC	NM_068122
*pmp-3*	ATACGAAGCCACGGATAG	CTGTGTCAATGTCGTGAAG	NM_001269679
*hsp-16.2*	GAGAGATATGGCTCTGATGG	TCTCCTTGGATTGATAGCG	NM_071106
*hsp-1*	GCACGGAAAGGTAGAAATC	CGAACTTGCGTCCAATAAG	NM_070667
*daf-16*	GAATGGATGGTCCAGAATG	GATTCCTTCCTGGCTTTG	NM_001026423
*G. gallus*			
*ACTB*	ATGAAGCCCAGAGCAAAAGA	GGGGTGTTGAAGGTCTCAAA	NM_205518
*PGK1*	GGATAAGGTGGATGTGAAGG	AGAACTTGTCAGGCATGG	NM_204985
*SDHA* [52]	CAGGGATGTAGTGTCTCGT	GGGAATAGGCTCCTTAGTG	NM_001277398
*YWHAZ*	AGAGTCGTCTCAAGTATCG	CAACCTCAGCCAAGTAAC	NM_001031343
*CLDN1*	GTCATGGTATGGCAACAG,	GGTGGGTAGGATGTTTCA	NM_001013611
*HSPA2*	GGCTGGAGAGAAGAATGT	GTGCTTACGCTTGAACTC	NM_001006685
*OCLN*	GCAGATGTCCAGCGGTTAC	GGTCCCAGTAGATGTTGGC	NM_205128

**Table 4 molecules-25-00835-t004:** Composition of the experimental diets (g/kg) and percentage of applied feed.

Ingredients	Experimental Period			
	1–21 d		22–42 d	
Maize	327.00	18.25%	329.66	18.37%
Wheat	314.60	17.56%	299.96	16.71%
Soybean meal (44% crude protein)	250.90	14.00%	265.00	14.77%
Soybean oil	50.80	2.84%	50.08	2.79%
Negative control premix or phytogenic premixes filled with a 1:1 mixture of wheat semolina and wheat bran to 5 g	5.00	0.28%	5.00	0.28%
Limestone	14.80	0.83%	14.60	0.81%
Monocalcium phosphate	14.00	0.78%	13.80	0.77%
Vitamin and mineral premix ^1^	12.00	0.67%	12.00	0.67%
Titanium dioxide	5.00	0.28%	5.00	0.28%
l-Lysine HCl	1.30	0.07%	2.30	0.13%
DL-Methionine	2.60	0.15%	2.00	0.11%
Threonine	2.00	0.11%	0.60	0.03%
Nutrient levels				
ME_N_ ^2^	12.48	0.70%	12.56	0.70%
Crude protein	220.8	12.32%	200.80	11.19%
Lysine	12.70	0.71%	12.00	0.67%
Methionine	5.90	0.33%	5.00	0.28%
Methionine/cysteine	9.70	0.54%	8.50	0.47%
Tryptophane	2.50	0.14%	2.30	0.13%
Threonine	8.60	0.48%	8.10	0.45%
Crude fibre	24.50	1.37%	24.00	1.34%
Crude fat	73.60	4.11%	73.90	4.12%
Starch	360.60	20.12%	390.80	21.78%
Sugars	42.90	2.39%	39.50	2.20%
Calcium	9.00	0.50%	8.80	0.49%
Total phosphorus	7.00	0.39%	6.70	0.37%
Sodium	1.60	0.09%	1.70	0.09%

^1^ The contents per kg of premix were as follows: 400,000 I.U. of vitamin A (acetate), 120,000 I.U. of vitamin D3, 8000 mg of vitamin E (α-tocopheryl acetate), 200 mg of vitamin K3, 250 mg of vitamin B1 (mononitrate), 420 mg of vitamin B2 (pure), 2500 mg of niacin (niacinamide), 400 mg of vitamin B6 (HCl), 2000 µg of vitamin B12, 25,000 µg of biotin (feed grade), 1000 mg of pantothenic acid (Ca d-pantothenate), 100 mg of folic acid (pentahydrate, feed grade), 80,000 mg of choline (chloride), 5000 mg of Zn (sulfate), 5000 mg of Fe (carbonate), 6000 mg of Mn (sulfate), 1,000 mg of Cu (sulfate) 20 mg of Se (sodium selenite), 45 mg of I (calcium iodate), 130 g of sodium (chloride), and 55 g of Mg (sulfate). ^2^ Values were estimated according to the WPSA equation (1984).

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
