# Peer review of "Ginseng Extract Ameliorates the Negative Physiological Effects of Heat Stress by Supporting Heat Shock Response and Improving Intestinal Barrier Integrity: Evidence from Studies with Heat-Stressed Caco-2 Cells, C. elegans and Growing Broilers"

_molecules, 2020, doi:10.3390/molecules25040835_

Round 1

Reviewer 1 Report

Dear Authors, 

Please see my comments below (major concern):

The experiment was based on 135 birds, my concern is why the number of sampling is too low (n=3 or n=6) especially error bars for some figures are huge, more analysis might reduce this error? Can you explain this? Otherwise, I think more analysis is required to support the results. My next comment is an example. Figure 4 and other figures. Variation for one of the bars is huge which makes difficult to understand your results. This makes your results invalid (either you had huge error in your analysis or lack of enough samples). Can you explain this?

Please see my comments below (minor concerns):

I highly recommended to follow the Journal's template, there are many formatting errors. 

In the text, please check for citations for figures (follow the journal's format). For example, Fig.1 should be Figure 1. Check all. Please provide figures or tables after where it explained, there is a gap between them.  Line 115. Please be consistent. Use at least one decimal for the numbers you provided in the text.  Formatting, there is no space between units and numbers for example Line 121. P values should be Italic.  Figure 3. You have error bars for control 45 which is good but for another control you have only for two of them. I would recommend you to only keep control 45 degree, except, you have an explanation for what you did. Please provide your reason if you did it intentionally. Figure 6. You used the same colour for bars. Please update it  Table 2. Composition of the experimental diets (g/kg). 1) is the title, the rest must be a caption, not the title. Also, this table needs to be more organised. You need to provide a total % of ingredients and there is a free space in mid of the table. Please fix this table. TEER unit, is it right? Subtitles all must be italic.  Line 478. Provide a reference for the statement in Line 477. Line 479. ad libitum should be italic.  Line 493-495. (Chickens showing any signs of illnesses, as indicated by evidence of diarrhoea, weight loss, mobility problems swollen joints, or other signs of illnesses, were removed from the experiment and culled). It's up to authors but I would recommend you remove this.  The method needs some info for example, the humidity of the environment, how to transport chicks from a hatchery. Please add more details about the method (section 4.3.1). Reference section. This is not the Journal's format as I checked. Please reformat the references. The template is available online. 

Regards, 

Author Response

We are grateful to the reviewer for reviewing our manuscript and his/her valuable feedback.

Please see my comments below (major concern):

The experiment was based on 135 birds, my concern is why the number of sampling is too low (n=3 or n=6) especially error bars for some figures are huge, more analysis might reduce this error? Can you explain this? Otherwise, I think more analysis is required to support the results. My next comment is an example. Figure 4 and other figures. Variation for one of the bars is huge which makes difficult to understand your results. This makes your results invalid (either you had huge error in your analysis or lack of enough samples). Can you explain this?

For a better understanding of our experimental approach and to circumvent misinterpretation, we would like to state a few points more precisely.

Non of the in vivo trials was based on the analysis of only 3 samples. There was only one Caco-2 experiment describing the effects of ginseng-extract on the epithelial barrier function for which 3 experiments had been performed. In the revised version of our manuscript, we extended the number of samples from 3 to 6. As indicated in Figure 3, we could unequivocally confirm the effects presented in the original draft, increasing the statistical significance. The number of samples selected from the broiler experiments was based on the calculated number of cases that had to be taken into consideration for the approval by the local state office of occupational health and technical safety (animal trial authority). In view of the proposed expected efficacy of the extract treatment, we were allowed to obtain tissue samples from a limited number of animals that were further analyzed by gene expression analysis presented in this study. In addition, some samples had to be excluded as the RNA quality did not pass our quality control (RIN value) ensuring high data integrity. Broilers that were only part of the performance trial were not killed but used for commercial breeding instead. This information was also added in the revised manuscript in the experimental section.    For a better characterization of the ginseng based effects, while keeping the number of live animals as low as possible, we extended our experimental approach by including in vitro and in vivo studies based on Caco-2 and C. elegans. Although the experimental variation in the broiler studies was elevated (but also partly significant!), the additional studies confirmed the basic effects observed in vivo.   

Please see my comments below (minor concerns):

I highly recommended to follow the Journal‘s template, there are many formatting errors. In the text, please check for citations for figures (follow the journal‘s format). For example, Fig.1 should be Figure 1. Check all. Please provide figures or tables after where it explained, there is a gap between them. Line 115. Please be consistent. Use at least one decimal for the numbers you provided in the text. Formatting, there is no space between units and numbers for example Line 121. P values should be Italic.

We have used the Journal’s template for initial submission. However, the reviewer is right, the draft included some formatting errors that have been addressed in the revised version. We changed the position of the figures in the document, the term “Figure” and the gene names (italic) throughout the manuscript. References were adapted according to the journal guidelines.

Figure 3. You have error bars for control 45 which is good but for another control you have only for two of them. I would recommend you to only keep control 45 degree, except, you have an explanation for what you did. Please provide your reason if you did it intentionally.

We have performed additional experiments that have been included in the revised manuscript to ensure statistical significance.

Figure 6. You used the same colour for bars. Please update it

Changed as requested.

Table 2. Composition of the experimental diets (g/kg). l) is the title, the rest must be a caption, not the title. Also, this table needs to be more organised. You need to provide a total °/o of ingredients and there is a free space in mid of the table. Please fix this table.

Table has been fixed and moved to the adequate position in the manuscript (now table 3).

TEER unit, is it right? Yes, Ohm is correct. Subtitles all must be italic. Changed as requested.  Line 478. Provide a reference for the statement in Line 477. Added as requested. Line 479. ad libitum should be italic. Changed as requested.  Line 493-495. (Chickens showing any signs of illnesses, as indicated by evidence of diarrhoea, weight loss, mobility problems swollen joints, or other signs of illnesses, were removed from the experiment and culled). It‘s up to authors but I would recommend you remove this. Changed as requested. The method needs some info for example, the humidity of the environment, how to transport chicks from a hatchery. Performance trial and slaughtering of selected broilers for gene expression analysis was performed under controlled conditions in the same facility without the necessity of relevant transport routes. Unfortunately, humidity data are not available. Please add more details about the method (section 4.3.1).

Reference section. This is not the Journal's format as I checked. Please reformat the references. The template is available online.

Changed as requested.  

Reviewer 2 Report

It is well documented that heat stress induces various physiological problems including immune dysfunctions, endocrine disturbances, respiratory alkalosis, and increased oxidative stress in poultry. 

It is incomprehensible for me to conduct this research on such different models. The goal of the research could be achieved on the chicken model. Intestinal barrier function is for example frequently determined by intestinal permeability (i.e.gut leakage) measured with electric resistance values for intestinal tissues and bloood LPS concentrations. The decreased intestinal barrier function is also often correlated with decreased expression of some tight junction-related genes and proteins in the GIT (OCLN, CLDN-1, and JAM-2). This possible effect could be correlated with chicken production parameters. Only than it could be concluded whether ginseng extract is a suitable feed additive in animal nutrition to reduce the negative physiological effects caused be heat stress.

The authors demonstrated a significant improvement in FCR (1-42d) in the ginseng extract group compared with the negative control group, and concluded that study clearly shows that ginseng extract increases the performance of heat stressed broiler chickens. It is not correct. This effect concerns only an improvement in FCR in the ginseng group compared with the negative control group. But the same effect was also observed before the ginseng extract application (1-21d).   

Three replicates per treatment considered for statistical evaluation in the growth study is below the standards, and do not guarantee reliability. 

Author Response

We thank the reviewer for reviewing our manuscript and his/her valuable feedback.

It is well documented that heat stress induces various physiological problems including immune dysfunctions,

endocrine disturbances, respiratory alkalosis, and increased oxidative stress in poultry. It is incomprehensible for me to conduct this research on such different model.

The goal of the research could be achieved on the chicken model. Intestinal barrier function is for example frequently determined by intestinal permeability (i.e. gut leakage) measured with electric resistance values for intestinal tissues and bloood LPS concentrations. The decreased intestinal barrier function is also often correlated with decreased expression of some tightjunction-related genes and proteins in the GIT (OCLN, CLDN-l, and JAM-2). This possible effect could be correlated with chicken production parameters. Only than it could be concluded whether ginseng extract is a suitable feed additive in animal nutrition to reduce the negative physiological effects caused be heat stress.

Our experimental approach including tissue samples from broilers in combination with in vitro and in vivo experiments was based on the following points:

The number of samples selected from the broiler experiments was based on the calculated number of cases that had to be taken into consideration for the approval by the local state office of occupational health and technical safety (animal trial authority). In view of the proposed expected efficacy of the extract treatment, we were allowed to obtain tissue samples from a limited number of animals that were further analyzed by gene expression analysis presented in this study. In addition, some samples had to be excluded as the RNA quality did not pass our quality control (RIN value) ensuring high data integrity. Broilers that were only part of the performance trial were not killed but used for commercial breeding instead. This information was also added in the revised manuscript in the experimental section.    For a better characterization of the ginseng based effects, while keeping the number of live animals as low as possible, we extended our experimental approach by including in vitro and in vivo studies based on Caco-2 and C. elegans. Although the experimental variation in the broiler studies was elevated (but also partly significant!), the additional studies confirmed the basic effects observed in vivo. We focused on the mentioned tight-junction proteins including OCLN and CLDN1 in the tissue samples obtained from broilers as well as the Caco-2 model (resulting in significant differences), which is a well-accepted system to characterize intestinal barrier function and study the expression of immune-regulatory, cell-cell adhesion and nutrient transport associated genes (doi: 4049/jimmunol.1700152; doi: 10.1007/s10565-005-0085-6). C. elegans was used as it represents a powerful system to study gene expression under heat shock conditions (e.g. doi: 10.1016/j.tig.2013.01.010). It is well established in our lab and generally accepted in the community to study molecular mechanisms in a eukaryotic organism. We believe it strengthens the results from our broiler studies and reduces the number of killed animals to the necessary minimum, as demanded by the trial authority.           

The authors demonstrated a significant improvement in FCR (1-42d) in the ginseng extract group compared with the negative control group and concluded that study clearly shows that ginseng extract increases the performance of heat stressed broiler chickens. It is not correct. This effect concerns only an improvement in FCR in the ginseng group compared with the negative control group. But the same effect was also observed before the ginseng extract application (1-21 d).

There was probably a misinterpretation regarding the trial implementation. The ginseng extract was applied during the whole period (1-42 d). Therefore, ginseng clearly shows a significant improvement in FCR. However, the trial also showed that ginseng application results in a better FCR under non-heatshock conditions. This information was also added in the table legend.

Three replicates per treatment considered for statistical evaluation in the growth study is below the standards and do not guarantee reliability.

Non of the in vivo trials was based on the analysis of only 3 samples. There was only one Caco-2 experiment describing the effects of ginseng-extract on the epithelial barrier function for which 3 experiments had been performed. In the revised version of our manuscript, we extended the number of samples from 3 to 6. As indicated in Figure 3, we could unequivocally confirm the effects presented in the original draft, increasing the statistical significance.

Reviewer 3 Report

The research behind this manuscript was well designed.

Minor comments

What is the reason behind to choose c. elegans for this study.? Because of Caco-2 cells – is enough supporting for broiler study. Any special reason.?

Revise abstract little more clarity to readers.

Line no. 75: cite with recent references (Eg. https://doi.org/10.3390/ani9080506)

Line no. 93: Caenorhabditis elegans

Line no. 174: remove “2.4. Figures and Tables.”

Avoid repeated sentences (Eg. Line no. 351, 405, 486) for ginseng extract expansion.

Author Response

The research behind this manuscript was well designed.

We thank the reviewer for acknowledging the quality of our study.

What is the reason behind to choose C. elegans for this study? Because of Caco-2 cells — is enough supporting

for broiler study. Any special reason?

Our experimental approach including tissue samples from broilers in combination with in vitro and in vivo experiments was based on the following points:

The number of samples selected from the broiler experiments was based on the calculated number of cases that had to be taken into consideration for the approval by the local state office of occupational health and technical safety (animal trial authority). In view of the proposed expected efficacy of the extract treatment, we were allowed to obtain tissue samples from a limited number of animals that were further analyzed by gene expression analysis presented in this study. In addition, some samples had to be excluded as the RNA quality did not pass our quality control (RIN value) ensuring high data integrity. Broilers that were only part of the performance trial were not killed but used for commercial breeding instead. This information was also added in the revised manuscript in the experimental section.    For a better characterization of the ginseng based effects, while keeping the number of live animals as low as possible, we extended our experimental approach by including in vitro and in vivo studies based on Caco-2 and C. elegans. Although the experimental variation in the broiler studies was elevated (but also partly significant!), the additional studies confirmed the basic effects observed in vivo. We focused on the mentioned tight-junction proteins including OCLN and CLDN1 in the tissue samples obtained from broilers as well as the Caco-2 model (resulting in significant differences), which is a well-accepted system to characterize intestinal barrier function and study the expression of immune-regulatory, cell-cell adhesion and nutrient transport associated genes (doi: 4049/jimmunol.1700152; doi: 10.1007/s10565-005-0085-6). C. elegans was used as it represents a powerful system to study gene expression under heat shock conditions (e.g. doi: 10.1016/j.tig.2013.01.010). It is well established in our lab and generally accepted in the community to study molecular mechanisms in a eukaryotic organism. We believe it strengthens the results from our broiler studies and reduces the number of killed animals to the necessary minimum, as demanded by the trial authority.           

Revise abstract little more clarity to readers. Line no. 75: cite with recent references (Eg. https://doi.org/10.3390/ani9080506).

Added as requested.

Line no. 93: Caenorhabditis elegans Line no. 174: remove “2.4. Figures and Tables.” Avoid repeated sentences (Eg. Line no. 351, 405, 486) for ginseng extract expansion.

Changed as requested.

Reviewer 4 Report

Reviewer has read this manuscript with great interest. This work reports that ginseng extract ameliorates the negative physiological effects of heat stress by supporting heat shock response and improving intestinal barrier integrity, which evidences were come from studies with heat-stressed Caco-2 cells, C. elegans and growing broilers. In this study, ginseng extract showed a beneficial influence in vitro, on the performance of growing broilers and on the heat resistance of C. elegans. Finally authors conclude that ginseng extract can be a useful feed ingredient to maintain the health and productivity of farm animals under environmental heat stress conditions. The experimental and theoretical methods described comprehensively. The research contents are clearly reported and the conclusions are supported by the data in this manuscript. The abstract is well matched with the text contents. The manuscript is organized well. Overall, the reviewer thinks that the subject should be of interest to the readers of Journal Molecules, and this manuscript is suitable for publication in this journal after addressing all comments

Comments and Suggestions for Authors

– In line 79, please delete "steroid glycosides and".because ginseng extract consists of triterpenoid saponins, not steroid glycosides.

– In line 83, please change "Ginsenoside-rich fermented" to "Ginsenoside of" because wild ginseng was fermented.

– Why hasn't the study of individual ginsenosides, example a lots of ginsenoside Re or ginsenoside Rb1, been carried out?

Author Response

Reviewer has read this manuscript with great interest. This work reports that ginseng extract ameliorates the negative physiological effects of heat stress by supporting heat shock response and improving intestinal barrier integrity, which evidences were come from studies with heat-stressed Caco-2 cells, C. elegans and growing broilers. In this study, ginseng extract showed a beneficial influence in vitro, on the performance of growing broilers and on the heat resistance of C. elegans. Finally authors conclude that ginseng extract can be a useful feed ingredient to maintain the health and productivity of farm animals under environmental heat stress conditions. The experimental and theoretical methods described comprehensively. The research contents are clearly reported and the conclusions are supported by the data in this manuscript. The abstract is well matched with the text contents. The manuscript is organized well. Overall, the reviewer thinks that the subject should be of interest to the readers of Journal Molecules, and this manuscript is suitable for publication in this journal after addressing all comments

We thank the reviewer for his/her valuable comments and for acknowledging the quality of our study.

Comments and Suggestions for Authors

In line 79, please delete "steroid glycosides and".because ginseng extract consists of triterpenoid saponins, not

steroid glycosides. In line 83, please change "Ginsenoside-rich fermented" to "Ginsenoside of" because wild ginseng was fermented.

Changed as requested.

Why hasn't the study of individual ginsenosides, example a lots of ginsenoside Re or ginsenoside Rbl, been carried out?

The general research question was more based on the characterization of the efficacy of a ginseng extract rather than focusing on single substances. In terms of application as a feed additive full extracts appear  more reasonable.

Round 2

Reviewer 1 Report

Dear Authors, Thank you for the revised version. Unfortunately, the P-values on error bars look not correct. You added P-values that show there are significant differences but the error bars are overlap (Figure 1, A and B), (Figure 2. C), (Figure 6.C). Still, numbers of sampling are not enough to support your results (variation for some results are high), figures 4 and 6. Provided reason (animal trial authority) is not a valid reason to have not enough samples for analysis. Unfortunately, there is not much improvement in the manuscript compared to the previous version, just some minor correction. The references still not the journal's format, abbreviations must be used for the name of journals in the reference section, not a full name. Regards,
